# Giant Magnetoimpedance Effect of Multilayered Thin Film Meanders Formed on Flexible Substrates

**DOI:** 10.3390/mi14051002

**Published:** 2023-05-06

**Authors:** Mengyu Liu, Zhenbao Wang, Ziqin Meng, Xuecheng Sun, Yong Huang, Yongbin Guo, Zhen Yang

**Affiliations:** 1School of Electronic and Information Engineering/School of Integrated Circuits, Guangxi Normal University, Guilin 541004, China; 1349375154@stu.gxnu.edu.cn (M.L.); 1354724210@stu.gxnu.edu.cn (Z.W.); meng@stu.gxnu.edu.cn (Z.M.); 2Key Laboratory of Integrated Circuits and Microsystems, Education Department of Guangxi Zhuang Autonomous Region, Guangxi Normal University, Guilin 541004, China; 3Guangxi Key Laboratory of Brain-Inspired Computing and Intelligent Chips, School of Electronic and Information Engineering, Guangxi Normal University, Guilin 541004, China; 4Microelectronic Research & Development Center, School of Mechatronics Engineering and Automation, Shanghai University, Shanghai 200444, China; sunxc@shu.edu.cn; 5Xidian-Wuhu Research Institute, Wuhu 241000, China; huangyong@xdwh-inst.com; 6Key Laboratory of UWB & THz of Shandong Academy of Sciences, Institute of Automation, Qilu University of Technology, Jinan 250014, China; guoyongbinecnu@163.com

**Keywords:** multilayered thin films, meanders, flexible substrates, giant magnetoimpedance effect, magnetic anisotropy

## Abstract

The giant magnetoimpedance effect of multilayered thin films under stress has great application prospects in magnetic sensing, but related studies are rarely reported. Therefore, the giant magnetoimpedance effects in multilayered thin film meanders under different stresses were thoroughly investigated. Firstly, multilayered FeNi/Cu/FeNi thin film meanders with the same thickness were manufactured on polyimide (PI) and polyester (PET) substrates by DC magnetron sputtering and MEMS technology. The characterization of meanders was analyzed by SEM, AFM, XRD, and VSM. The results show that multilayered thin film meanders on flexible substrates also have the advantages of good density, high crystallinity, and excellent soft magnetic properties. Then, we observed the giant magnetoimpedance effect under tensile and compressive stresses. The results show that the application of longitudinal compressive stress increases the transverse anisotropy and enhances the GMI effect of multilayered thin film meanders, while the application of longitudinal tensile stress yields the opposite result. The results provide novel solutions for the fabrication of more stable and flexible giant magnetoimpedance sensors, as well as for the development of stress sensors.

## 1. Introduction

Lately, multilayered thin film meanders in the application of giant magnetoimpedance (GMI) effect sensors have become a research hot topic. Firstly, multilayered thin film has higher GMI effect compared with single-layer thin film [1,2]. Secondly, meander configurations can further enhance the GMI effect and utilize the space of the line element to improve the spatial resolution [3]. Finally, thin film can be miniaturized and developed by micro-electro-mechanical (MEMS) technology such as magnetron sputtering and micro-electroplating, and integrated with circuits easily. Recent observation of multilayered thin film meanders has tended to focus on biomedical sensing [4,5,6,7], nondestructive testing [8,9], and pressure detection [10] based on the ultrasensitive and easy-to-implement GMI effect.

The magnetoimpedance of soft magnetic thin film is not only sensitive to external magnetic fields but also sensitive to strain or pressure. Stress-impedance effects in multilayered FeSiB/Cu/FeSiB and FeCuNbCrSiB/Cu/FeCuNbCrSiB thin film meanders were studied [11,12], in which the thin film meanders were developed on a silicon cantilever by MEMS technology. In addition, Wang et al. systematically fabricated sandwiched Cr/Cu/NiFe/Cu/NiFe multilayered thin film meanders on a glass substrate by MEMS technology and investigated the pressure-impedance effect under the vertical mechanical pressures produced by a self-made pressure-sensing system [10]. However, conventional thin film GMI elements fabricated on rigid substrates such as silicon and glass cannot meet the requirement for ease of installation on an irregular plane or one with a large curvature [13]. Flexibility of the substrate is a necessary feature for some applications. Flexible substrates normally are more rugged, lighter, portable, higher in thermal and chemical stability, and less expensive to manufacture compared to their rigid substrate counterparts. Due to the advantages of flexible substrates, the potential impacts of flexible magnetoelectronic devices on physiological health detection, human–computer interaction, and touch displays have received increasing attention [14,15,16,17].

To date, some interesting results on flexible GMI sensors with strip-line multilayered structures have been reported by distinct research groups. Nanostructured FeNi-based multilayered thin films were formed on cyclo-olefin copolymer (COC) and cyclo-olefin polymer (COP) substrate and used for low-pressure sensing. The structure, magnetic properties, and GMI were comparatively analyzed [18,19,20,21]. Li et al. reported a multilayered FeNi/Cu/FeNi film developed on Kapton substrate; a special flexible microstrip transmission line was employed to investigate the GMI sensor’s magnetic field and frequency response and dependence on the sensor’s deflection [22]. Recently, Jin et al. fabricated CoFeSiB films on different flexible substrates and explored the influence of stress on GMI performance [23]. Thin films deposited on polymeric substrates generally show significant magnetic anisotropy after deposition. There has been an increasing focus on the link between magnetic properties and flexible systems [24]. Dynamic magnetic response was investigated through the GMI of flexible NiFe/Ta and FeCuNbSiB/Ta multilayers under external stress [25,26].

So far, multilayered thin film meanders have usually been deposited by sputtering on a nonflexible substrate. In general, multilayered meander configurations, in comparison with single-layer ones, have low stored stress, enabling in some cases the control of magnetic properties. Meanwhile, research has verified the impact of magnetron configurations on ferromagnetic thin films on flexible substrates in relation to high-frequency applications and magneto-transport properties [27,28]. In this sense, studies considering meander structures and flexible substrates are of interest to make easy the integration of these samples as sensor elements in GMI-based flexible magnetoelectronic devices. Flexible magnetic microfluidic systems are also a very attractive application field [29]. Among flexible substrates, polyimide (PI) and polyethylene terephthalate (PET) are often used to replace traditional rigid substrates due to their advantages of good folding resistance, high light transmission, and smooth surfaces.

Therefore, in this work, multilayered FeNi/Cu/FeNi thin film meanders are prepared by DC magnetron sputtering and MEMS technology on PI and PET flexible substrates, respectively. The magnetic properties of multilayered thin films on different substrates are comparatively investigated. Arched and concave 3D-printed holders are used for fixing the samples and exerting tensile and compressive stresses, which can be converted from tensile to compressive strains by adjusting the curvature of the free edge of the holders from arched to concave. The effects of different tensile and compressive stresses on the GMI of multilayered thin film meanders are systematically analyzed.

## 2. Experimental Details

In this context, we choose PI and PET as flexible substrates to prepare multilayered thin film meanders, and a glass substrate was also set up as a control test. The meanders were designed as three-turn structures, with the ‘n’-shaped meander part denominated as a “one-turn” sample. The meander element consists of Ti/FeNi/Cu/FeNi multilayers. The multilayered thin film meander samples fabricated on glass, PI, and PET substrates were denominated as S1, S2, and S3, respectively. The whole length and width of the sensing elements are 5 and 2.3 mm, respectively. The FeNi and Cu line widths are each 300 µm, and the line spacing is 100 µm. The thicknesses of the FeNi magnetic monolayer and Cu conductive layer are 1 µm and 0.9 µm, respectively. The design of the geometry of the meanders is shown in Figure 1b. The multilayered thin film meanders were fabricated by MEMS technology as shown in Figure 1a. Before fabrication, the substrate was ultrasonically cleaned in acetone, alcohol, and deionized water in turn for 10 min, and then dried with nitrogen. The objective of the cleaning is to remove the impurities from the surface of the substrate and improve its adhesion to the film. The detailed fabrication steps are as follows: (ⅰ) spin positive photoresist with a thickness of 5 μm on glass and flexible substrate, bake it on the hot plate at a temperature of 363.15 K for 4 min, and then cool it down to room temperature; (ⅱ) pattern the photoresist film by UV exposure and development in a mixture solution for 80 s; (ⅲ) deposit Ti, FeNi, and Cu layers in turn by DC magnetron sputtering with a base pressure below 1.5 × 10^−6^ torr and apply a constant 150 Oe bias magnetic field during the sputtering to induce a transverse anisotropy in the magnetic layer, with the argon pressure and sputtering power being 3 × 10^−3^ torr and 150 W, respectively, and the Ti layer being used as buffer layer; (ⅳ) remove the unpatterned photoresist by lift-off technology and obtain the multilayered thin film meanders; (ⅴ) bend the sample for testing. Figure 1c,d show the fabricated meanders on PI and PET substrates, respectively.

We specify that when the meanders are on an arched surface, the meanders are subjected to tensile stress (marked as positive), and when the meanders are on a concave surface, the meanders are subjected to compressive stress (marked as negative). To effectively apply tensile and compressive stresses, we designed three arched and concave holders with different heights and curvatures for stress transfer to multilayered thin film meanders. The holders were fabricated using 3D printing technology using photopolymer resin as the material and used for adhering the samples as shown in Figure 2, where h denotes the height of the stress device and R denotes the radius of curvature. The tensile and compressive stresses applied to the meanders changed with the height of the holders. Table 1 describes the magnitudes of stresses with different substrates and holders.

An Alpha-Step-D-300 probe step gauge from KLA was used to measure the thicknesses of samples. A field-emission scanning electron microscope (FESEM, CARL-ZEISS SMT) operating at an accelerating voltage of 2 kV was used to investigate the surface morphology of multilayered thin film meanders. Structural analysis of the deposited multilayered thin film meanders was performed using an X-ray diffractometer (XRD, Miniflex 600) with a scan range of 30°–90° and a fixed scan rate of 2°/min. The surface roughness of the deposited FeNi films was studied using an atomic force microscope (AFM, NX20). Hysteresis loops were measured at room temperature with a vibrating sample magnetometer (VSM, Quantum Design SQUID-VSM (MPMS-3)).

The GMI effect of the samples was tested using a 4294A impedance analyzer (Keysight) with an alligator clip and the 16,048 G accessory. The AC frequency was from 100 Hz to 120 MHz. Previous studies have demonstrated that altering the current amplitude of the AC band can significantly affect the output response of the GMI sensor [30]. In order to ensure that the applied mechanical stress was the sole experimental variable, we standardized the AC current amplitude at a fixed value of 10 mA [31]. The external magnetic field was supplied by adjustable Helmholtz coils, and the direction of the applied field extended in the direction of the long axis of the sample (longitudinal). The magnetic field strength of −110 Oe to 110 Oe was controlled by changing the gas gap between the Helmholtz coils and direct current produced by a power supply. The measurement schematic is shown in Figure 3.

## 3. Results and Discussion

### 3.1. Characterization of Multilayered Thin Film Meanders

Due to the poor thermal conductivity of the flexible substrates, it takes longer time to deposit a multilayered sample with the same thickness on a flexible substrate, as the sample is locally overheated during the deposition process, leading to inhibition of the growth of the film. The SEM surface morphologies of samples with the same thickness prepared on different substrates were observed as showed in Figure 4a–c. It is evident that the multilayered thin film meanders adhered well to the surfaces of the different substrates and produced smooth and uniform thin films. However, as is evident from these images, the substrates have a significant impact on the surface morphology of the multilayered thin film meanders. The magnetic thin films on flexible substrates display nucleation clusters and multiple cracks, as seen in Figure 4b,c. The phenomenon may be derived from the lower thermal conductivity and higher internal stress of the flexible substrates [32]. The microscopic morphology of the multilayered thin film meanders was characterized by AFM, as shown in Figure 4e,f. The scanning area was 10 µm × 10 µm on different substrates. It was found that the multilayered thin film meanders on the hard glass substrate show a small root mean square (RMS) value of 3.42 nm; however there were distinct peaks and valleys that were both sharper and steeper. Conversely, the samples on the PI and PET substrates had higher RMS values of 7.61 nm and 12.86 nm, respectively. It is worth noting that the microstructure of samples on PET substrate was smooth and continuous, without any distinct sharp peaks or valleys. Therefore, multilayered thin film meanders with PET as a flexible substrate also have excellent magnetic properties.

Figure 5 displays the XRD patterns of multilayered thin films grown on different substrates. The smaller the grain size of the crystal, the more diffuse and broadened the diffraction pattern becomes. Therefore, within a certain range, larger crystal grain size results in a wider range of activity of magnetic domains and better magnetic properties. It can be clearly seen that the particle sizes of the multilayer thin films on different substrates are similar. Among all samples, the characteristic FeNi and Cu (111) diffraction peaks are obvious at around 2θ ≈ 44°, indicating the preferred crystal growth orientation of the face-centered cubic (fcc) FeNi film along the (111) plane. The second maximum of the characteristic diffraction peak at around 2θ ≈ 51° corresponds to the reflection (200) of FeNi fcc material. Fixed grain boundaries have an impact on the structure of magnetic domains, limiting the movement or rotation of magnetic domain walls within the grain. Additionally, the peak positions of thin films deposited on flexible substrates shifted slightly, which is consistent with previous reports [20]. Based on the diffraction intensity, the crystallinity of the samples on PI and PET substrates is poor, at 0.74 and 0.71, respectively, which is lower than that of the samples on hard glass (0.85). Research shows that the higher the crystal quality, the lower the proportion of defects. As mentioned above, this is consistent with conclusions drawn from SEM and AFM analysis.

Figure 6 shows the normalized hysteresis loops of the multilayered thin film meanders deposited on the different substrates at room temperature. The magnetic field is applied to the easy axis orientation (long axis) of the multilayered thin film meanders. It was found that the samples deposited on different substrates all exhibit typical ferromagnetism with good soft magnetic properties. In addition, the samples on different substrates have a low-saturation magnetization field (~125 Oe), which can provide the GMI effect with a high magnetic sensitivity in the applied magnetic field (−110 Oe to 110 Oe). The multilayered thin film meanders exhibit significant magnetic anisotropy on both rigid and flexible substrates, owing to the external magnetic field applied during the deposition process [33]. Additionally, because of the stresses within the flexible substrate itself, the samples deposited on flexible substrates have higher magnetic anisotropy values. To further compare the performances of the substrates, the coercivity was analyzed on different substrates. The inset shows that the multilayered thin film meanders with a rigid substrate and the samples on a flexible substrate have relatively similar low coercivity (2–6 Oe). This suggests that multilayered thin film meanders on flexible substrates possess good soft magnetic properties. The results are consistent with the AFM and SEM analysis.

### 3.2. GMI Measurements of Multilayered Thin Film Meanders under Bending Loads

#### 3.2.1. Stress Estimation of Meanders on Different Flexible Substrates

We conducted a 3D nonlinear finite element analysis (FEM) of the multilayered thin film meanders’ stacking configuration using the COMSOL software based on plane strain conditions. Table 2 lists the relevant physical parameters and thicknesses of the materials used in the finite element analysis. Figure 7 depicts the stress distribution in meanders under convex and concave loading models on PI and PET substrates. The simulation results show that the stresses are uniformly distributed over each microstrip line. Moreover, at the same curvature, multilayered thin film meanders deposited on PI substrates are subjected to greater stress than those deposited on PET substrates because the Young’s modulus and thickness of the PI substrate are greater than those of PET. The stress σ exerted by the multilayered thin films on a flexible substrate is calculated using the Stoney equation, which is defined as follows:(1)$$\sigma =\frac{Y}{6R(1-V_{s})}\frac{t_{s}}{t_{fm}}$$
where *Y* represents the Young’s modulus of the substrate, *V_s_* denotes the Poisson’s ratio of the substrate, *R* signifies the radius of curvature of the substrate, *t_s_* represents the thickness of the substrate, and *t_fm_* stands for the thickness of the material.

#### 3.2.2. Giant Magnetoimpedance Effect under Stress

The stress can also induce the rearrangement of domain walls, reorientation of magnetic domains, and modifications of circumferential magnetic anisotropy, resulting in diverse magnetostrictive properties and permeability, which affect the skin depth and GMI effects [34,35]. The Jiles–Sablik model can describe the effect of applied stress on magnetic permeability, which can be expressed in Equation (2):(2)$$H_{e}=H_{ex}+\alpha M+\frac{3\lambda_{s}\sigma M}{2\mu_{0}M_{S}^{2}}$$
where *H_ex_* is the applied magnetic field, *μ*_0_ is the magnetic permeability in a vacuum, *λ_s_* is the saturation magnetostriction constant, *M* is the magnetization intensity, *M_s_* is the saturation magnetization intensity, *α* is the dimensionless average field of coupling between domains, and *H_e_* and H are both vector quantities. The effective magnetic field (*H_e_*) of the sample is strongly influenced by the anisotropic field generated from the mechanical stress applied to the sample. Magnetoimpedance curves provide a more convenient way to illustrate the changes of magnetic anisotropy in thin films. Therefore, to better demonstrate the correlation between applied stress and GMI, the GMI ratio is defined in Equation (3):(3)$$GMI(\%)=\frac{Z(H)-Z(H_{\max})}{Z(H_{\max})}\times 100\%$$
where *Z*(*H*) refers to the impedance value in a specific external magnetic field, and *Z*(*H_max_*) refers to the impedance when the maximum external magnetic field is applied, signifying that the multilayer film has reached magnetic saturation. Sensitivity is one of the important parameters of sensor performance and is given by Equation (4):(4)$$\xi =\frac{d(\Delta Z/Z)}{d(H)}$$
where *d*(Δ*Z*/*Z*) represents the disparity between the peak value of (Δ*Z*/*Z*) and the value of (Δ*Z*/*Z*) when the applied magnetic field is at its maximum level. Similarly, *d*(*H*) represents the corresponding value of the applied magnetic field when the GMI ratio reaches its peak value.

Figure 8a,b show the field dependence of GMI responses at f = 23 MHz under the different stress states for meanders on PI and PET substrates, respectively. The GMI curves exhibit similar trends under different substrates and stress conditions. The GMI ratio of the meanders increased first and then decreased as Hex increased. This can be explained in terms of the magnetization rotation model [36]. In the entire magnetic field range, the GMI response curve shows a clear double-peak shape with two maximum values of the positive and negative external magnetic field approaching the magnetic anisotropic field, which is a typical feature of the GMI effect in soft magnetic thin films with transverse magnetic anisotropy. The maximum external magnetic field is insufficient to magnetically saturate the sample. Thus, the GMI ratio of the sample within the largest curvatures reaches its maximum negative value at zero field and gradually increases with increasing applied field. The largest negative GMI ratio could be attributed to the changes in the magnetization mechanism and the inductance of the meanders caused by the stresses. Previous studies have shown that tensile and bending loads lead to significant changes in the inductance of soft magnetic materials, and the magnetic induction strength tends to decrease with increasing tensile and compressive stress, leading to changes in the impedance of materials [37]. In addition, the GMI ratio of a thin film with almost zero magnetostriction is usually positive [23].

Figure 8a shows the trend of the sample on the PI substrate with different applied stresses. The maximum distinguished GMI ratio under different stresses occurs near the external field, close to the magnetic anisotropic field, defined as peak field H_k_. When no mechanical stress is applied, the H_k_ of the samples on PI substrate is 18.1 Oe with a maximum GMI ratio of 39.5%. With the compressive stress increases, the peak field of the samples on the flexible substrate gradually shifts to the higher fields, and the GMI response monotonically increases. When the external compressive stress reaches 260 MPa (at a holder height of 1.7 mm), the H_k_ of the sample shifts 13 Oe, and the maximum GMI ratio increases by 29.5%. This phenomenon may be attributed to the close relationship between the H_k_ of the sample and the magnetic anisotropic field, where longitudinal compression stress leads to an increase in the transverse anisotropic field [22]. In addition, the deformations caused by stresses affect the self-inductance and mutual inductance between the strip-line of the meanders, changing the impedance of the sample and thereby affecting the GMI response. Conversely, with increasing tensile stress, the H_k_ of the sample gradually shifts to a lower field, and the GMI response rapidly decreases. It is worth noting that when the tensile stress is equal to 260 MPa, the peak field of the sample reaches zero field, exhibiting the typical characteristics of high longitudinal magnetic anisotropy of thin films. Moreover, the GMI ratio under the maximum tensile stress is close to zero. Under the higher stress, more cracks were found from micro-observation, resulting in higher impedance and a lower GMI effect. The GMI curves of the samples on PET substrates and PI substrates showed the same trend under stress variation as shown in Figure 8b. However, accompanying the stress changes, the peak field of the samples on PET substrates only shift slightly (maximum shift of 5.2 Oe at 32 MPa of compression stress), and the change trend of the GMI ratio is not significant compared with that of the PI substrates.

Figure 9 demonstrates the maximum GMI response of samples under different stresses. With the increase of tensile stress, the maximum GMI ratio gradually decreases, while the maximum GMI ratio gradually increases with the increase of compression stress. At a holder height of 1.7 mm, the maximum GMI ratios observed for the samples on PI and PET substrates were 69% and 48% with sensitivities of 2.2/Oe and 1.9/Oe, respectively. Notably, under the same curvature conditions, multilayered thin film meanders on PI substrates exhibited higher sensitivity to stress/strain changes in GMI response compared to those on PET substrates. As a consequence, PI is an ideal material as a substrate for pressure and strain sensors, while PET is more suitable for the preparation of flexible GMI sensors with more stable magnetic properties.

## 4. Conclusions

In this paper, FeNi/Cu/FeNi multilayered thin film meanders were fabricated on hard glass, PET, and PI substrates. By comparing the surface roughness, crystal structure, and magnetic properties of the films on different substrates, it was found that the multilayered thin film meanders deposited on flexible substrates have the same advantages of good density, high crystallinity, and excellent soft magnetic properties compared with those deposited rigid substrates. The effects of different tensile and compressive stresses on the GMI effect of the multilayered thin film meanders were analyzed comparatively. The results indicate that the GMI ratio increased with increasing compressive stress and decreased with increased tensile stress, and the sensitivity of multilayered thin film meanders presented a similar variation trend. Additionally, the giant magnetoimpedance effect under stress is more significantly affected by the PI substrate than the PET substrate at the same curvature. Therefore, multilayered thin film meanders on PI substrates are more sensitive to stress, which makes them suitable substrates for stress sensor devices. PET substrates are better suited for producing stable and flexible giant magnetoimpedance sensors, providing a new solution for achieving biomolecular detection in flexible microfluidic systems.

## Figures and Tables

**Figure 1 micromachines-14-01002-f001:**
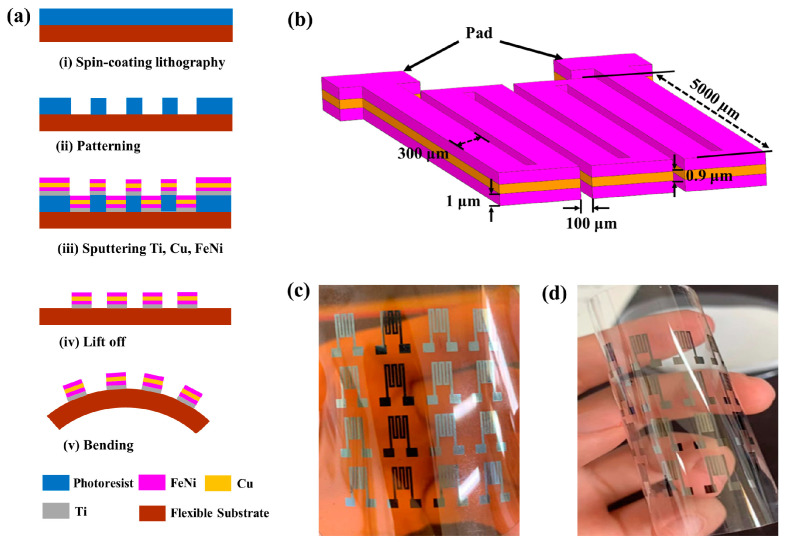
MEMS fabrication steps for the meanders (**a**); the design schematic diagram of meanders (**b**); fabricated meanders on PI substrate (**c**) and PET substrate (**d**).

**Figure 2 micromachines-14-01002-f002:**
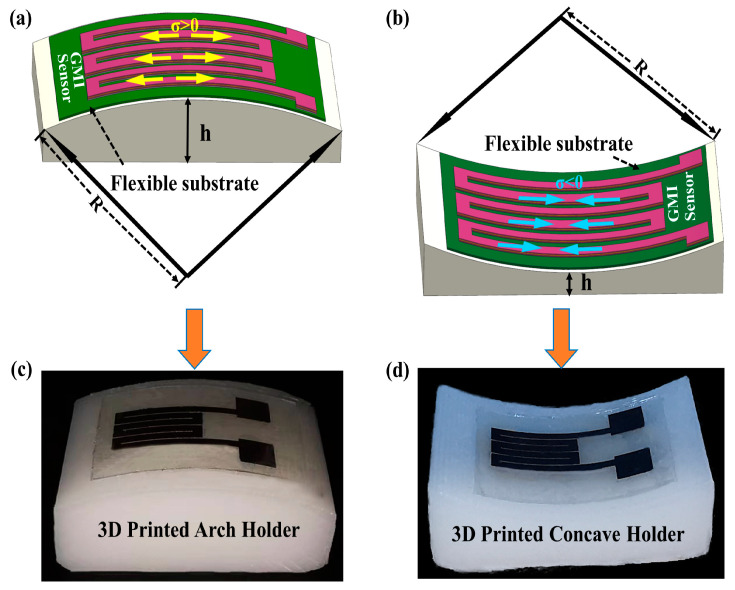
Sketch of tensile strain (**a**) and compressive strain (**b**) applied to the samples; the fabricated meanders on 3D-printed arched holder (**c**) and concave holder (**d**).

**Figure 3 micromachines-14-01002-f003:**
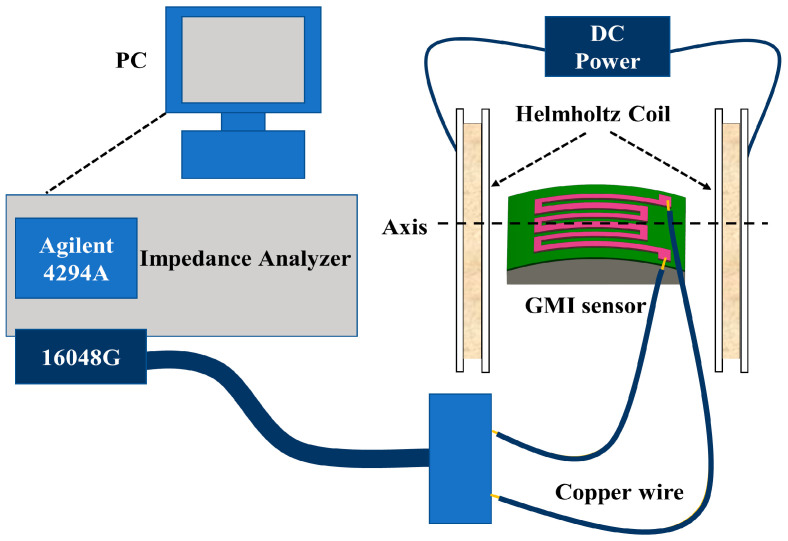
The schematic diagram of magnetoimpedance measurement under stress.

**Figure 4 micromachines-14-01002-f004:**
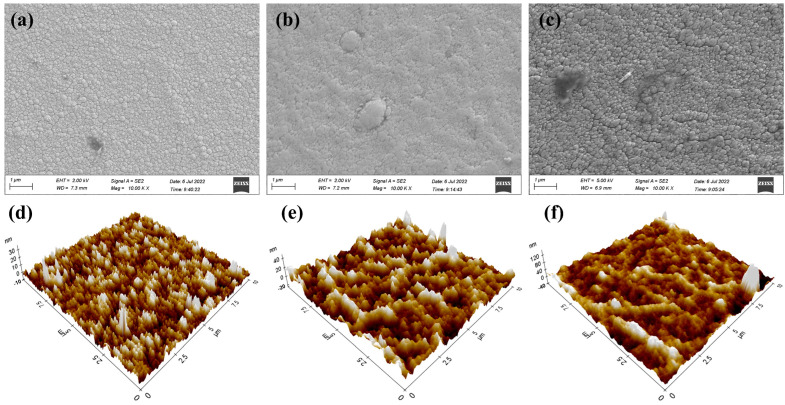
SEM and AFM images of multilayered thin film meanders deposited on different substrates: hard glass (**a**,**d**); PI (**b**,**e**); PET (**c**,**f**).

**Figure 5 micromachines-14-01002-f005:**
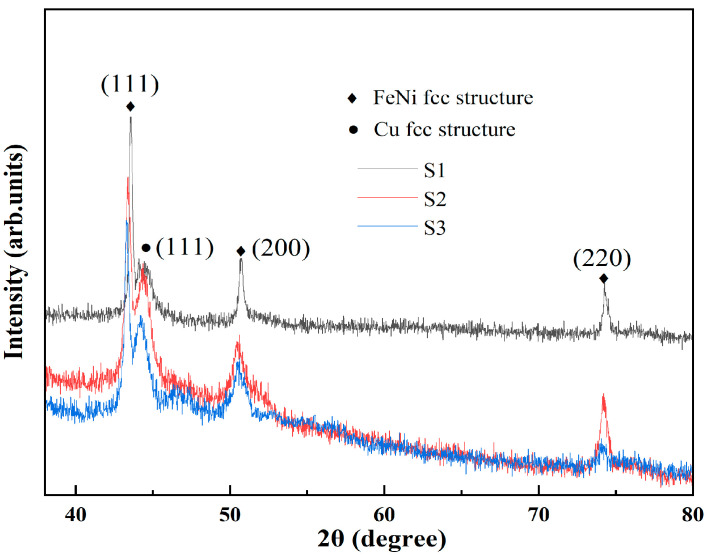
X-ray diffraction patterns of multilayered thin film meanders on different substrates.

**Figure 6 micromachines-14-01002-f006:**
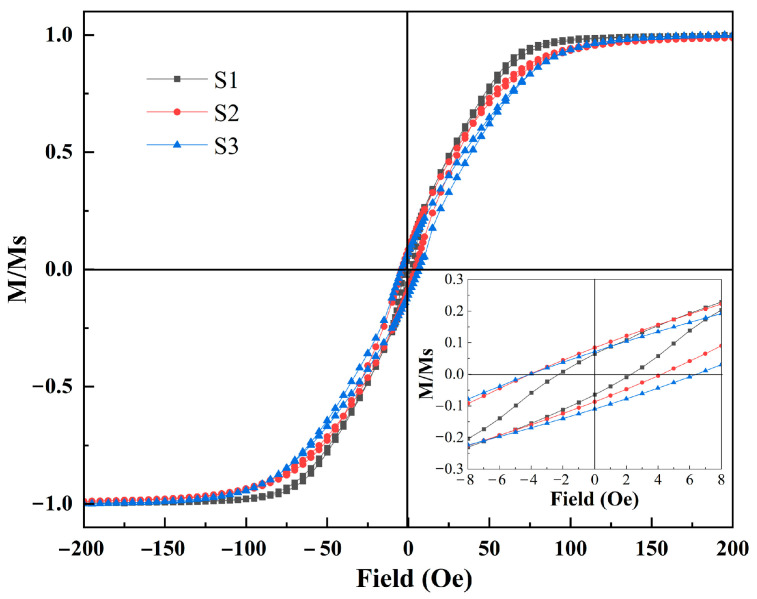
Hysteresis loops of multilayered thin film meanders on different substrates.

**Figure 7 micromachines-14-01002-f007:**
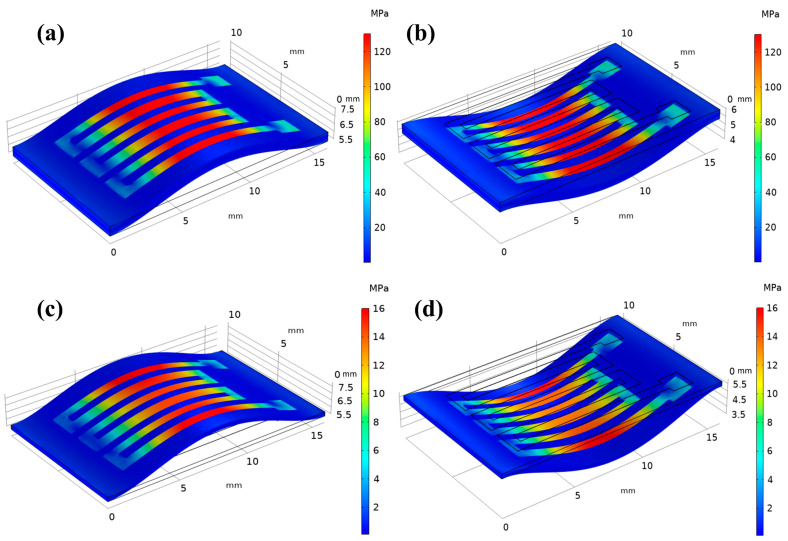
Simulated stress distributions applied to the multilayered thin film meanders: tensile (**a**) and compressive stress (**b**) at 130 MPa applied to the PI substrate; (**c**) tensile and (**d**) compressive stress at 16 MPa applied to the PET substrate.

**Figure 8 micromachines-14-01002-f008:**
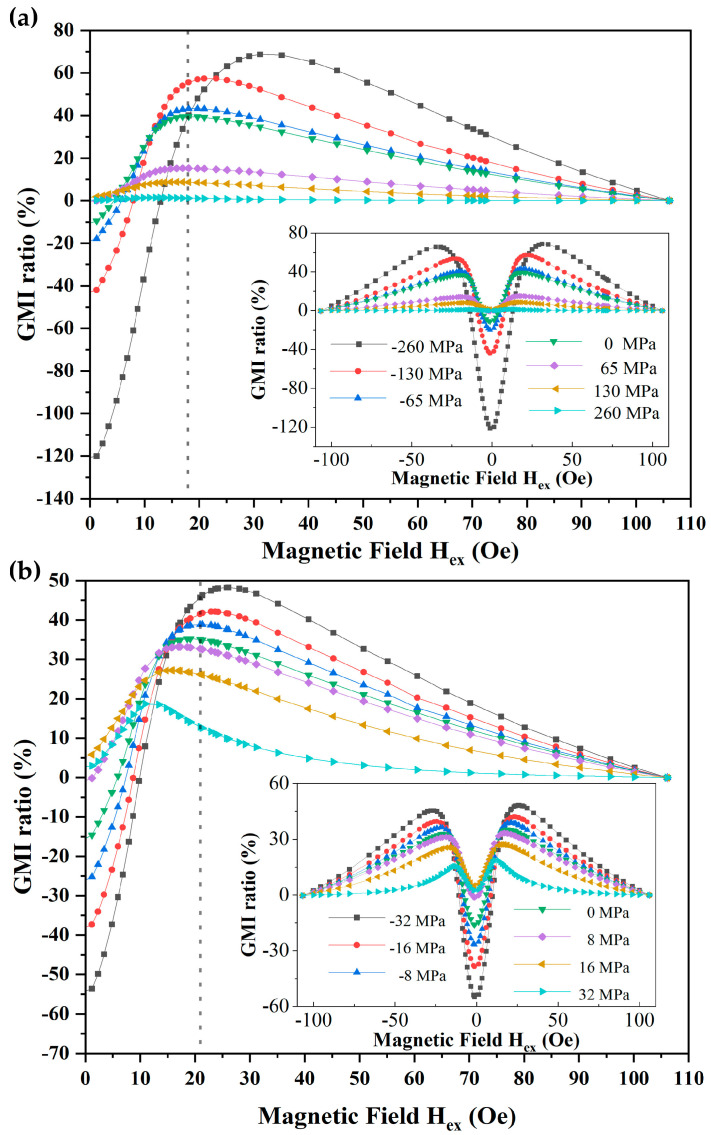
GMI response curves for multilayered thin film meanders with excitation frequency of 23 MHz at different stresses for the substrates: PI (**a**) and PET (**b**).

**Figure 9 micromachines-14-01002-f009:**
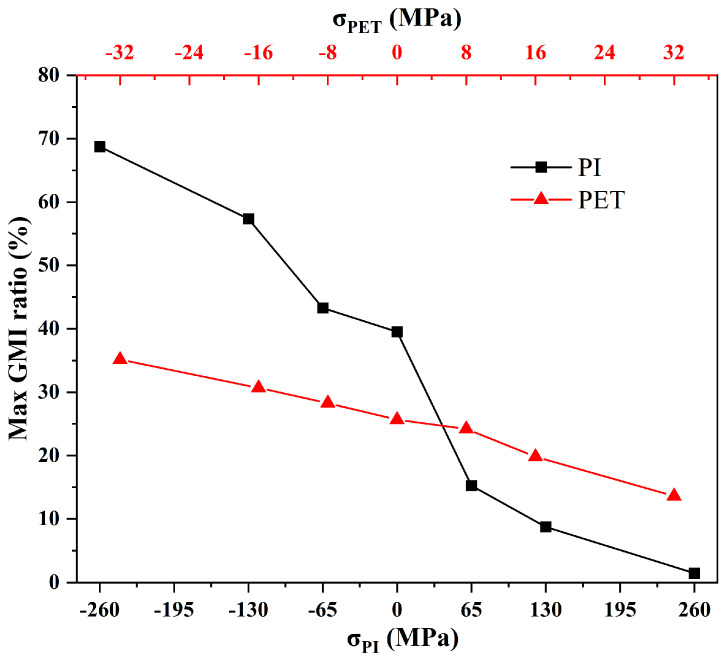
Maximum GMI ratios of multilayered thin film meanders on PI and PET substrates at different stresses.

**Table 1 micromachines-14-01002-t001:** Relationship between the height of the holders (h) and the stress on the PI and PET substrates.

h (mm)	σ_PI_ (MPa)	σ_PET_ (MPa)
1.7	−260	−32
3.1	−130	−16
4.1	−65	−8
5	0	0
5.9	65	8
6.9	130	16
8.3	260	32

**Table 2 micromachines-14-01002-t002:** The parameters of the materials used in FEM analysis.

Material	Poisson’s	Young’s Modulus (×10^3^ MPa)	Thickness (μm)
Adhesive layer	0.3	4.5	30
FeNi film	0.31	129	1
Cu film	0.32	117	0.9
PI substrate	0.3	4.8	300 (experiment)
PET substrate	0.37	2.9	100 (experiment)

## Data Availability

Not applicable.

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
