# Peer review of "Giant Magnetoimpedance Effect of Multilayered Thin Film Meanders Formed on Flexible Substrates"

_micromachines, 2023, doi:10.3390/mi14051002_

Round 1

Reviewer 1 Report

See attached file.

Reviewer 2 Report

Please, make the following corrections

1. References citation in the first paragraph of section 1 is bold, but in the rest of the paper, these are in normal style.

2. In reference section, the bullets are not arranged.

Author Response

  1. References citation in the first paragraph of section 1 is bold, but in the rest of the paper, these are in normal style.

Reply: thank you very much for this important comment.

We have removed the bolding from the reference citations to bring them into line with the later text. (Please see the revised manuscript at the page 1).

  1. In reference section, the bullets are not arranged.

Reply: thank you very much for this important comment.

We have modified the arrangement of the bullets in the reference section and removed the spaces before the numbers. (Please see the revised manuscript at the page 12-13).

Reviewer 3 Report

In order to have a valid scientific analysis of the magnetic samples when GMI is measured, it is very important to specify the measurement current and this current has to be of constant amplitude. The current generates a circular magnetic field when passing through a conductor, so affecting the magnetic response and behavior of the sample. So, to get a significant measurement it is necessary to measure with a constant current amplitude, and to analyze its influence to measure at different current amplitudes. Please clarify if the 4294A allows the measurement at constant current amplitude, and could be very interesting to make the measurements at different current amplitudes, as important changes in GMI are expected (you can see the paper JMMM 378(2015)485–492 to see this important effect).
